# Clustering and Kernel Density Estimation for Assessment of Measurable Residual Disease by Flow Cytometry

**DOI:** 10.3390/diagnostics10050317

**Published:** 2020-05-18

**Authors:** Hugues Jacqmin, Bernard Chatelain, Quentin Louveaux, Philippe Jacqmin, Jean-Michel Dogné, Carlos Graux, François Mullier

**Affiliations:** 1Hematology Laboratory, NAmur Research Institute for LIfe Sciences (NARILIS), Namur Thrombosis and Hemostasis Center (NTHC), CHU UCL Namur, Université catholique de Louvain, 5530 Yvoir, Belgium; Bernard.chatelain@gmail.com (B.C.); francois.mullier@uclouvain.be (F.M.); 2Montefiore Institute, University of Liege, 4000 Liège, Belgium; q.louveaux@uliege.be; 3MnS–Modelling and Simulation, 5500 Dinant, Belgium; philippe.jacqmin@m-n-s.be; 4Pharmacy Department, University of Namur, 5000 Namur, Belgium; jean-michel.dogne@unamur.be; 5Department of Hematology, Namur Research Institute for Life Sciences (NARILIS), Namur Thrombosis and Hemostasis Center (NTHC), CHU UCL Namur, Université catholique de Louvain, 5530 Yvoir, Belgium; carlos.graux@uclouvain.be

**Keywords:** acute myeloid leukemia (AML), flow cytometry, multiparametric data analysis, clustering, kernel density estimation, personalized medicine

## Abstract

Standardization, data mining techniques, and comparison to normality are changing the landscape of multiparameter flow cytometry in clinical hematology. On the basis of these principles, a strategy was developed for measurable residual disease (MRD) assessment. Herein, suspicious cell clusters are first identified at diagnosis using a clustering algorithm. Subsequently, automated multidimensional spaces, named “Clouds”, are created around these clusters on the basis of density calculations. This step identifies the immunophenotypic pattern of the suspicious cell clusters. Thereafter, using reference samples, the “Abnormality Ratio” (AR) of each Cloud is calculated, and major malignant Clouds are retained, known as “Leukemic Clouds” (L-Clouds). In follow-up samples, MRD is identified when more cells fall into a patient’s L-Cloud compared to reference samples (AR concept). This workflow was applied on simulated data and real-life leukemia flow cytometry data. On simulated data, strong patient-dependent positive correlation (*R*^2^ = 1) was observed between the AR and spiked-in leukemia cells. On real patient data, AR kinetics was in line with the clinical evolution for five out of six patients. In conclusion, we present a convenient flow cytometry data analysis approach for the follow-up of hematological malignancies. Further evaluation and validation on more patient samples and different flow cytometry panels is required before implementation in clinical practice.

## 1. Introduction

Multiparameter flow cytometry (MFC) is a powerful technology for cell phenotyping, capable of analyzing multiple parameters on millions of single cells in a short period of time [1]. This technique is helpful for the diagnosis of hematological malignancies, and its use for disease monitoring has gained a large amount of interest in the last decades through the evaluation of minimal/measurable residual disease (MRD). However, the pitfalls of current MRD assessment strategies by MFC are still numerous, with pre-analytical, analytical, and post-analytical issues. It has now become obvious that the use of MFC for this purpose requires some degree of standardization/harmonization and less subjectivity [2,3,4,5].

The immunophenotypic follow-up of acute myeloid leukemia (AML) is a difficult topic, mainly given the important heterogeneity of the disease [6,7,8]. Multiple approaches such as the follow-up of leukemia-associated immunophenotypes (LAIP) and “different from normal” (DfN) strategies have been proposed [9,10]. The European LeukemiaNet (ELN) working party published a consensus document on AML MRD combining the two strategies into a “LAIP-based DfN approach” [11]. Nevertheless, no data of this combined approach have yet been published [12]. 

In this study, we aimed to propose an original and more reproducible analysis strategy for the development of promising MFC biomarkers for disease-based MRD assessment. This strategy was applied to data from AML patients for creating patient-specific MRD models.

For creating good models, the performance of a model must be optimized in regard to its objective. The objective of the present approach was to obtain a strategy that confers a very high positive predictive value (PPV) for relapse. Importantly, the interest of a model with high PPV would take the position of a first-line classification tool for MRD—positive tests would not require further testing for evaluating disease persistence, whereas negative tests would require further investigations.

## 2. Experimental Section

### 2.1. Study Material

This study received the approval of the ethics committee of the University Hospital, CHU UCL Namur, Belgium (CE-164/2018)). Retrospective bone marrow MFC data from 6 AML patients at diagnosis (6 FCS (flow cytometry standard) files) and follow-up (46 FCS files) were used, as well as 20 reference patients (control group), thus yielding a total of 72 FCS files, from which an additional 36 simulated MRD FCS files were generated. All FCS files are available in open source in the Appendix A section. Patient characteristics and flow diagram are given in Table 1 and Figure 1. The reference samples included nine bone marrows of immune/idiopathic thrombocytopenic purpura (ITP) patients, eight “normal bone marrows”, and three bone marrows from patients with underlying solid cancer without bone marrow infiltration. The “normal bone marrows” were drawn from patients with unexplained mild cytopenia. Follow-up of these patients always concluded on transient or constitutive mild cytopenia without clear explanation and without malignant evolution. For all samples, the same in-house standardized 10-parameter MFC assay was performed using a FACS Canto II (BD Biosciences, Erembodegem, Belgium). The panel used comprised antibodies to CD45, CD3, CD13, CD19, CD33, CD34, CD117 and HLA-DR. Forward and side scatter features (FSC and SSC, respectively) were also taken into account for analyses. For this algorithm, every recorded cytometry event was considered as a cell, in order to avoid potential subjectivity through the manual elimination of debris or doublets. In this specific study, this was considered acceptable because there was no evidence of major differences in the amount of debris/doublets in the patient cohort compared to the reference sample cohort.

### 2.2. Study Methods

The algorithm presented in this study was developed using the Infinicyt software (Cytognos, Salamanca, Spain, V2.0.1.c.100). Two data analysis methods were used consecutively: clustering and contour gating. In the Infinicyt software, clustering is based on cell density and a *k*-nearest neighbor (KNN)-based algorithm using the Euclidean distances of transformed raw data (patent no. US10133962B2). For contour gating, kernel density estimation is used through the use of the “reference image” tool of the Infinicyt software. It allows delineating the position of a clustered cell population in two-dimensional spaces on the basis of density calculations. In this study, the largest possible kernel density estimation functions available were used (probability model with a resolution of 64 bins) according to the Infinicyt software.

On the basis of the consecutive use of these two data analysis methods, three new concepts were described: the “Cloud”, the “Leukemic Cloud” or “L-Cloud”, and the “Abnormality Ratio” or “AR” (Figure 2). Briefly, a Cloud represents a model describing the position of a cell cluster (obtained by a clustering algorithm) in the multidimensional space. The L-Cloud is a Cloud for which the cell cluster is predominantly malignant; it is identified at diagnosis of a hematological malignancy. Of note, multiple L-Clouds may be identified at diagnosis. Both Clouds and L-Clouds are assay-specific and patient-specific. The AR is a calculated parameter obtained by dividing the percentage of patient cells falling into an (L-)Cloud by the percentage of cells from a control group (reference samples) falling into the same (L-)Cloud.

The following workflow (AR/L-Cloud workflow) was then designed (Figure 2):(a)From a diagnostic AML sample, clustering is used on MFC data to obtain cell clusters.(b)Suspicious cell cluster(s) are then identified on the basis of their aberrant immunophenotypic profile. These are usually CD45 low, with low SSC [13], and are often CD34-positive and form in most cases a unique cluster of >10% of total cells. However, given the important heterogeneity of AML, a suspicious cell cluster should always be identified by experienced cytometrists on the basis of their scientific knowledge of the disease.(c)Once suspicious cell clusters are identified, “Cloud(s)” are created. A “Cloud” is created by the Boolean intersection of the contour gates of 45 bi-parametric plots (each parameter vs. each parameter, using logicle transformation for fluorescence parameters and linear transformation for FSC and SSC).(d)Once a Cloud is created at diagnosis, its Abnormality Ratio (AR) can be calculated. The AR is calculated as follows: AR = ((Cloud cells/total cells) of patient sample)/((Cloud cells/total cells) of control group sample). Note: If # Cloud cells = 0, use 1.(e)The Cloud with the highest AR, formed by a cell cluster of at least 1 × 10^4^ cells (arbitrary value) at diagnosis will define a “Leukemic Cloud” or “L-Cloud”. Of note, if multiple Clouds of at least 5 × 10^3^ cells have an AR of >1000 (arbitrary value), all should be considered as L-Clouds. The Cloud with the highest AR and at least 1 × 10^4^ cells will be considered the “the major L-Cloud”. MRD assessment at follow-up will be done through the AR calculation at follow-up of L-Clouds.

#### 2.2.1. Endpoint 1: Theoretical Evaluation of the AR/L-Cloud Concept

For the theoretical evaluation, the “L-Cloud specificity” was defined by the number of cells of the reference samples falling outside an L-Cloud divided by the number of cells of the reference samples analyzed. “L-Cloud sensitivity” was defined by the number of cells from the “L-Cloud cell cluster”* falling into the L-Cloud divided by the number of cells contained into this same cluster.(** Cell cluster used to define the L-Cloud*).

##### Global Evaluation of the AR/L-Cloud Concept

For this endpoint, simulated MRD MFC data were created by computationally spiking AML cells into 1 × 10^6^ cells of reference samples.

Obtaining AML cells

Using the diagnostic FCS files of the six AML patients and the AR/L-Cloud workflow detailed above, 1 × 10^4^ cells were sampled (by bootstrapping in R (R-3.5.2)) out of the cell cluster defining the major L-Cloud of each AML patient. We thus obtained six FCS files, each containing 1 × 10^4^ cells from the “major L-Cloud cell cluster”* (AML-FCS). (**Cell cluster used to define the major L-Cloud*).

Simulation of MRD samples

First, the FCS files of the 20 reference samples were merged, and then 1 × 10^6^ cells were sampled (by bootstrapping in R (R-3.5.2)) and saved in a separate FCS file. We thus obtained one FCS file with a mixture of 1 million normal cells (NORM-FCS). Thereafter, simulated MRD was created by computationally spiking a decreasing number of randomly selected cells from the “major L-Cloud cell cluster”* (5000, 1000, 500, 100, 50, 10 cells) to the NORM-FCS file, that is, ≈0.5%, 0.1%, 0.05%, 0.01%, 0.005%, and 0.001% simulated MRD. All these simulated follow-up data were saved into separate FCS files, thus obtaining 36 FCS files. (**These cells were different than the cells of the AML-FCS files*).

Evaluation procedure

L-Clouds were created using the AML-FCS files.Performance characteristics (specificity and sensitivity) of the different L-Clouds were calculated using the AML-FCS files and the NORM-FCS file.The AR was calculated for each of the 36 simulated follow-up data and compared to the expected theoretical results for each patient.

##### Evaluation of the Influence of Each Measured Parameter on the Intrinsic Performance of the L-Cloud

For this endpoint, the AML-FCS files and the NORM-FCS file were used.

Evaluation procedure

Using the Infinicyt software and the L-Clouds (based on the AML-FCS files), a forward selection was performed on the basis of the specificity value (using the NORM-FCS file) for all steps except the first one. First, the contour gate of the bi-parametric FSC-A/SSC-A graph was selected; the contour gates related to the other parameters were then introduced one at a time. At each step, the improvement in specificity was calculated for each remaining parameter and the one giving the highest improvement was selected. The process was repeated until all parameters were introduced. At this stage, the L-Cloud was not optimized in terms of sensitivity/specificity—no criteria were used to limit the inclusion of parameters and no backwards deletion was performed.

#### 2.2.2. Endpoint 2: Clinical Evaluation of the AR/L-Cloud Concept

For this endpoint, the diagnostic and follow-up MFC data of the 6 AML patients were used, as well as all the data from the 20 reference patients. Using the proposed AR/L-Cloud workflow, AR was calculated at diagnosis and for each follow-up time point. AR results were also evaluated with regard to clinical aspects (treatments) and compared with morphological (remission/blast percentage) and molecular information when available.

## 3. Results

Results related to the theoretical evaluation of the AR/Cloud concepts (endpoint 1) are given in Table 2 and Figure 3. Table 2 provides the results with regard to the in silico simulated MRD data (endpoint 1a) with all AR and the number of cells falling into the L-Cloud for each simulation. Correlation coefficients between AR and expected theoretical results showed a perfect correlation for each patient (*R*^2^ = 1). The mean sensitivity for the L-Clouds was 66.28%, whereas the specificity ranged from 99.9502% (patient 5) to 99.9999% (patient 4). The specificity had a major impact on AR outcome, ranging for example from 2.38 (patient 5) to 720.28 (patient 4) for the 0.1% MRD simulation.

Figure 3 shows the evaluation of the influence of each measured parameter on the intrinsic performance of each L-Cloud (endpoint 1b). This highlights between-patient differences with regard to the added value of each parameter. Of note, for all patients in this study, the addition of CD19 and CD3 reduced the L-Cloud sensitivity without impacting the L-Cloud specificity.

Figure 4 shows the results related to the clinical evaluation of the AR/L-Cloud concepts. It represents real-life data. The total number of cells analyzed for each patient and reference patient are summarized in Appendix A (Appendix A). Appendix A also includes the number of cells retrieved into the L-Clouds for each study patient and reference patient.

For patient 1, a decrease of AR was observed after induction therapy, yet remained >5, although morphological remission was obtained and MLL fusion transcripts became undetectable. At follow-up 6, 2 months after allogeneic stem cell transplantation (ASCT), the AR was 15, which can be considered quite high after curative therapy. At follow-up 7, relapse was characterized by 35% of blasts in morphological assessment and an AR of 463.2.

For patient 2, AR also declined after therapy. The 1 log reduction of AR between follow-ups 3 and 4 while the patient was off-treatment may have represented long-lasting effects of chemotherapy or recovered anti-tumor immunity, although none of this was demonstrated. After ASCT, an AR < 1 was observed for the two major malignant cell clusters identified at diagnosis in this patient (L-Cloud 1: AR1, and L-Cloud 2: AR2).

For patient 3, a sharp decrease in AR was observed after induction therapy. AR was < 1 at the second follow-up. An increase of AR was, however, observed at follow-ups 3 and 4, before decreasing again after the initiation of anti-Wilm’s tumor 1 (WT1) vaccination.

For patient 4, a decline in AR until follow-up 4 was observed. After ASCT, the AR fluctuated. For patients 3 and 4, the AR increase before consolidation therapy (AR > 20) may have contributed to the hypothesis that consolidation therapy (anti-WT1 vaccination for patient 3 and ASCT for patient 4) was essential to the patients’ respective treatments.

For patient 5, the AR decreased after the first induction therapy and became <1 after the second induction therapy. The AR << 1 could be explained by the aplastic status of the bone marrow as assessed by morphological examination.

For patient 6, a decrease in AR was observed and became < 1 at follow-up 3, although, again, the patient was off-therapy between follow-ups 2 and 3. AR increased to 14 at follow-up 6 in spite of 2 induction cures, which appeared quite high. Molecular relapse was objectivized at follow-up 7 through an increase of nucleophosmin 1 (NPM1) transcripts. At follow-up 8, morphological relapse was frank, with 96% of blasts. However, AR remained low, suggesting the emergence of a different clone (from an immunophenotypical point of view) than that of the initial diagnosis. This observation may question the involvement of the consolidation therapy in this “immunophenotypic shift”.

## 4. Discussion

MRD monitoring through MFC is challenging, especially in AML, and obtaining consensus about how to proceed is difficult [5,8,11,14,15]. In this proof of concept pilot study, we validated an original data analysis strategy for the development of future MFC MRD biomarkers.

The strategy presented in this study began with an unsupervised data analysis approach (clustering), thereby identifying cell clusters. The position of these clusters in the multidimensional space could then be modeled. We chose a non-parametric approach based on a combination of two-dimensional kernel density estimation models. This choice was made to avoid the use of 10-dimensional black box data analysis approaches (deep learning, neural networks, etc.), as these latter approaches do not allow easy expert supervision. The created models were dubbed as “Clouds”. In order to avoid time-consuming efforts, not every cell cluster needed to be modeled as a Cloud (although possible), but only suspicious cell clusters, chosen on the basis of expert selection. From these models/Clouds, the most interesting for MRD assessment were selected (L-Clouds). The AR concept that we developed was obtained by dividing the number of patient cells (as percentage of total patient cells) falling into a Cloud, by the number of cells from a control group/reference samples (as percentage of total control group cells) falling into the same Cloud. Departing from diagnostic patient samples, Clouds with the highest AR and formed by cell clusters of at least 1 × 10^4^ cells were tagged as “Leukemic Clouds” or “L-Clouds” (see Section 2.2). This cut-off of 1 × 10^4^ cells was set arbitrarily and may be optimized in future studies, as well as the AR cut-off of 5 × 10^3^ cells for secondary L-Clouds with AR > 1000. The AR follow-up of L-Clouds was then proposed as methodology for MRD assessment. In this regard, proposals for AR interpretation at diagnosis could be made—a very high AR (> 10,000–100,000) at diagnosis indicates that the AR could be very useful for MRD assessment. A moderate AR (1000–10,000) indicates that the AR could be moderately useful for follow-up, whereas low AR (< 1000) indicates that the AR will be of limited use for follow-up. 

A first point highlighted by the AR is that the higher the AR at diagnosis, the higher the specificity of the L-Cloud, and the lower the amount of normal (= non-malignant) cells into this L-Cloud. In this study, important differences in the L-Cloud specificities were observed. This means that some L-Clouds define a multidimensional region where almost no normal cells were found (patients 3 and 4), whereas other L-Clouds are barely specific and identify a region where many normal cells are found (patients 5 and 6). Specificity in this regard is synonymous of “the degree of aberrancy in regard to normality”, with high specificity meaning high degree of aberrancy. The AR is the “clinically user-friendly” translation of this specificity. It can be rapidly and easily quantified at diagnosis. A lack of specificity of an L-Cloud could be improved by using more immunological markers in extended mutliparametric flow cytometry assays to identify more aberrations. However, over-extended multiparametric flow cytometry assays (> 10/12 colors) would probably be counterproductive with regard to standardization (for the moment). For future application, we could therefore recommend performing multiple well-developed standardized multiparametric assays at diagnosis and selecting the one for follow-up that highlights malignant cell clusters with high specificity on the basis of the AR concept. On the basis of these elements, the “perfect L-Cloud” would correspond to an “empty space” for all reference control samples (= 100% specificity) and would encompass all malignant cells for a given patient. However, according to the results of this study, “perfect L-Clouds” do not exist, which is in line with the theory of leukemia-associated immunophenotypes (LAIPs) showing variable specificities and sensitivities with respect to normality for each LAIP [14]. 

The second notable element about the AR is that an AR of 1 means that there are, proportionally, as many cells falling into the L-Cloud from a patient sample as the mean of reference samples. An AR of 2 means that there are two times more cells from a patient sample falling into the L-Cloud than the mean of reference samples. Given the interindividual variability among reference samples (see Appendix A), an AR above 1 does not mean necessarily that there is disease persistence. Although this pilot study did not allow for the identification of guidelines for AR interpretation at follow-up to obtain a very high positive predictive value for relapse (the main objective of this approach), this should be determined in future studies. Two main strategies can be investigated for AR interpretation. Firstly, AR cut-offs as clinical trigger can be studied. In this regard, cut-offs may be subdivided with regard to different AR categories at diagnosis. Secondly, an increase in AR as clinical trigger can be studied. If AR increases significantly over a defined period of time, this would mean there is disease evolution with higher risk for relapse. However, in our opinion, the use of multivariate statistics (multiple logistic regressions) or artificial intelligence-based algorithms (e.g., random forest) on multiple standardized biomarkers would probably be the best approach to eventually better define the risk of relapse. 

The comparison of the AR/L-Cloud approach with other already published flow cytometry MRD approaches is summarized in Table 3. The AR/L-Cloud approach is the most similar to the LAIP approach, as both are diagnostic-based. However, due to the well-known subjectivity linked to the LAIP approach [14], it has been decided not to assess MRD in the AML patients through this approach. 

### 4.1. Limitations of This Study

Regarding the limitations of the proposed strategy, the lack of sensitivity for detecting MRD must be considered and will be linked to multiple well-known causes [8,12,16,17,18]. These major causes are the lack of standardization, immunophenotypic shifts (clonal selection, therapy-related, interference-related, etc.), and sometimes limited aberrancies of the malignant cell population identified at diagnosis on the basis of the parameters studied. Although the lack of assay standardization must be controlled, sensitivity issues linked to immunophenotypic shifts and low aberrancies are more difficult to resolve and will need the development/use of complementary biomarkers to detect MRD, as highlighted by the results of patient 6 in this study. For this patient, a gain of CD13 and HLA-DR expression was observed for the malignant cell population at relapse (data not shown), which led to a different L-Cloud than at diagnosis. Interestingly, discrepancies between AR results and other biomarkers such as morphology and genetics should be considered and could be used to detect possible clinically relevant immunophenotypic shifts at relapse [16,19,20].

Another limitation is the choice of the reference control samples, which is an important part of the proposed strategy that impacts the specificity of the L-Cloud and thereby the sensitivity for MRD detection. In this study, the reference samples were chosen on the basis of the absence of evidence of underlying disease. Such reference samples should include “normal” and “regenerative” bone marrow samples. Ideally, international guidelines regarding the use of “reference bone marrow samples” should be established and/or commercial assays must be developed and validated using a fixed “reference database” that is unique for each technically and clinically validated assay.

Finally, this study is a monocentric pilot study based on retrospective data, with all well-known limitations associated with retrospective design. Further prospective, adequately powered studies are required to better evaluate the potential of the proposed methodology.

### 4.2. Perspectives of This Study

This study is a pilot study underpowered for a validation of the concept. Additional optimization work is still to be done. For example, we did not evaluate the variability among experts for the identification of the suspicious cell clusters at diagnosis. Although this step remains partly subjective among experts, improving objectivity in this field is possible. We used fixed clustering and contour gating algorithms of the Infinicyt software and did not compare them to other clustering and contour gating methods (e.g., three-dimensional kernel density estimation). Moreover, the proposed classification algorithm has yet not been compared towards more conventional classification algorithms (support vector machine, random forest, neural networks, ensemble learning etc.). However, as mentioned before, such algorithms are often considered “black box algorithms”, hampering their use in clinical practice, as they do not allow easy biological supervision. Continuous improvement in assay standardization and increasing the amount of cells to be acquired needs to also be part of further evaluation.

Interestingly, the AR concept could also be developed for a “follow-up-based DfN approach”, (in contrast to a diagnostic-based DfN approach) where clustering analysis is performed directly on follow-up samples. Clouds may then be created for each suspicious cell cluster at follow-up. AR of the Clouds can then be calculated, and analogically to L-Clouds, AR thresholds for these Clouds could be defined as predictors of relapse in clinical studies. Although probably complementary to diagnostic-based approaches, more difficulties will probably be linked to this kind of DfN approach: (a) normality appears more difficult to define (than pathology); (b) a large amount of cells must be acquired to obtain significant cell clusters, allowing the creation of significant Clouds; and (c) such an approach would require more computational power.

## 5. Conclusions

This study highlighted the major interest of implementing new data analysis concepts and tools in clinical hematology MFC [21,22,23]. The strategy developed here relies on the principles of standardization, multiparametric data analysis, and personalized medicine. Further improvement and evaluation of this strategy through method comparison, as well as retrospective and prospective studies on multiple MFC panel designs is required to further explore its potential applicability in clinical practice.

## Figures and Tables

**Figure 1 diagnostics-10-00317-f001:**
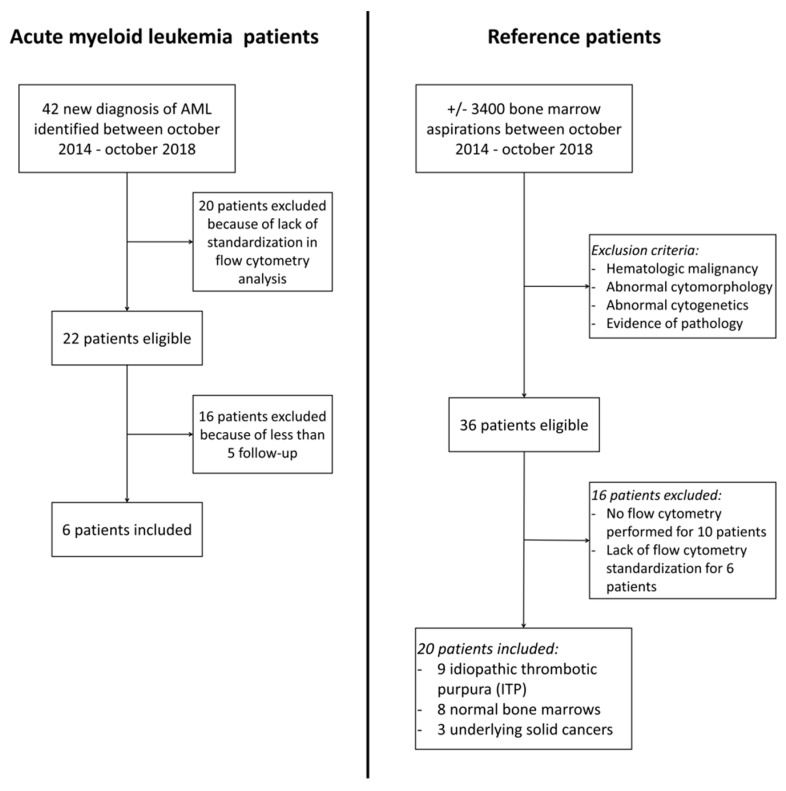
Flow diagram of acute myeloid leukemia patients and reference patients.

**Figure 2 diagnostics-10-00317-f002:**
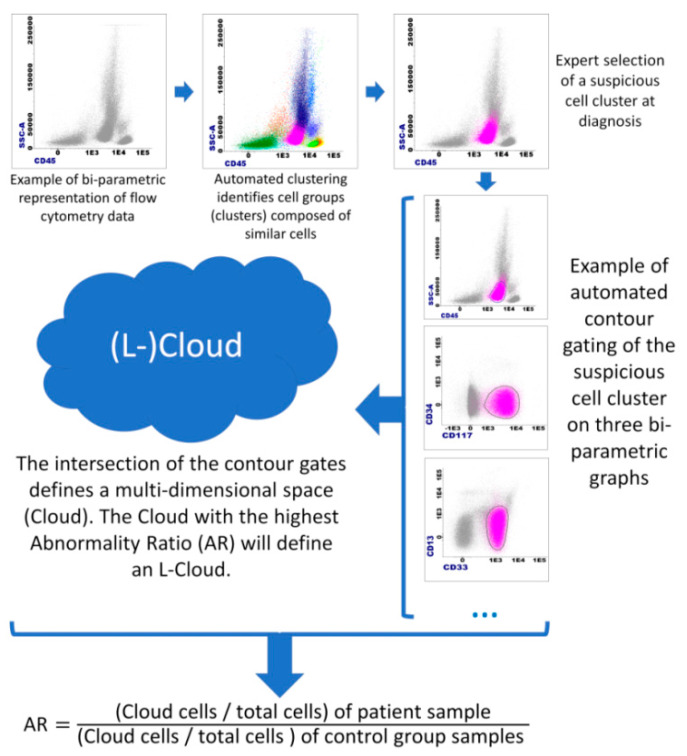
The principle of the “Cloud”, “L-Cloud”, and “Abnormality Ratio” concepts; AR: Abnormality Ratio. Infinicyt software (Cytognos, Salamanca, Spain) was used for flow cytometry data representation, automated clustering (*k*-nearest neighbor-based clustering), and automated contour gating (two-dimensional kernel density estimation).

**Figure 3 diagnostics-10-00317-f003:**
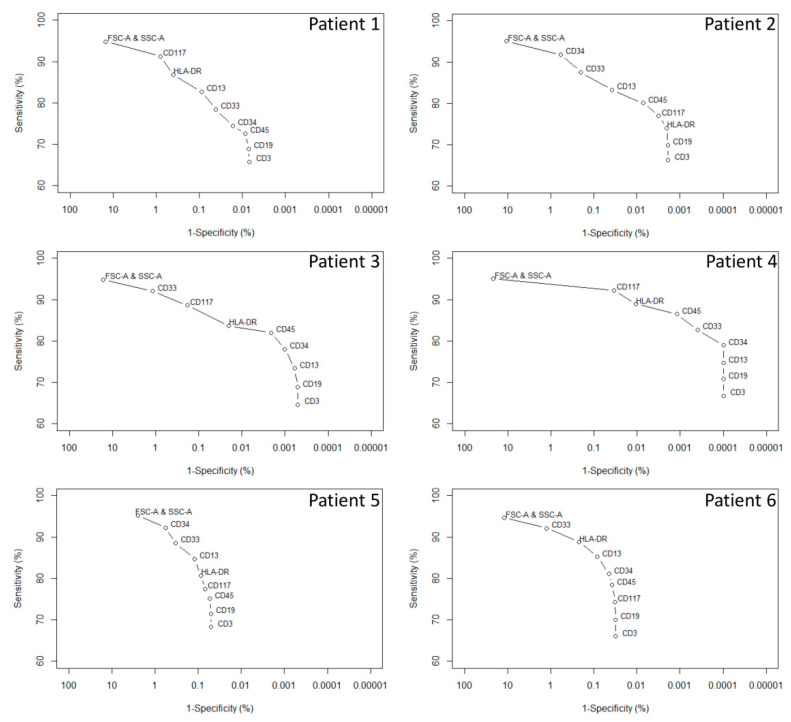
Influence of the parameters on the sensitivity and specificity of each patient-specific L-Cloud. The abscissa (1-specificty %) indicates the false positive rate of the L-Cloud (%); the ordinate indicates the L-Cloud sensitivity (%). Using Infinicyt software and patient-specific L-Clouds, established on the basis of 10^4^ cells from the major malignant cell cluster of six AML patients at diagnosis (patients 1–6), a forward selection based on the L-Cloud specificity (using a mixture 1 × 10^6^ of normal cells from reference samples) was performed for each patient (1–6). In a first step, the FSC-A and SSC-A parameters were selected. In the following steps, the other parameters were introduced one at a time. At each step, the improvement in specificity was determined for each remaining parameter and the one with the highest improvement was added. The process was repeated until all parameters were introduced. The L-Cloud represents a model describing a multidimensional space where malignant cells are found. This region is patient-specific and is established following a well-defined algorithm at diagnosis (see Section 2.2). The L-Cloud sensitivity is defined by the number of cells from the “L-Cloud cell cluster” falling into the L-Cloud divided by the number of cells contained into this same cluster (= 10^4^ cells) as a percentage. The L-Cloud specificity is defined by the number of cells of the control group falling outside the L-Cloud divided by the number of cells of the control group analyzed (= 10^6^ cells) as a percentage.

**Figure 4 diagnostics-10-00317-f004:**
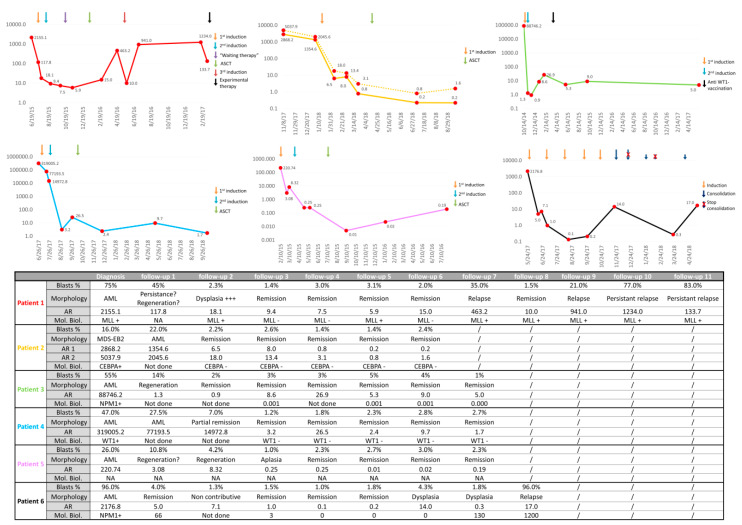
Comparison of the Abnormality Ratio (AR) with morphology and molecular biology results over time for six acute myeloid leukemia patients. Graphics: abscissa = time (DD/MM/YY); ordinate = AR; ASCT: allogeneic stem cell transplantation; induction: induction therapy; consolidation: consolidation therapy; Blasts%: percentage of blast cells among nucleated cells established on morphological basis on bone marrow aspirate; Morphology: bone marrow aspiration conclusion based on morphological results; Mol. Biol.: bone marrow aspirate molecular biology results (follow-up columns) based on mutations at diagnosis (diagnosis column). The presence of MLL at follow-up was evaluated by RT-PCR, with a limit of detection (LOD) of 5%. The presence of CEBPA mutation at follow-up results was evaluated by PCR, followed by Sanger sequencing, with a LOD of 10%. The presence of NPM1 at follow-up was evaluated by RT-PCR with a limit of detection of 1:10,000 cells. NPM1 results are expressed in terms of percentage ratio (NPM1/ABL1); the presence of WT1 expression at follow-up was evaluated by RT-PCR (+ = WT1 overexpression; − = no WT1 overexpression). RT-PCR: reverse transcriptase–polymerase chain reaction; NPM1: nucleophosmin 1; MLL: partial tandem duplication of KMT2A gene; CEBPA: CCAAT enhancer binding protein alpha gene; WT1: Wilm’s tumor 1 gene; AR: Abnormality Ratio. AR1 and AR2 for patient 2 were defined because two cell clusters of at least 5 × 10^3^ cells had an AR > 1000 at diagnosis (two different L-Clouds).

**Table 1 diagnostics-10-00317-t001:** Clinical, cytogenetical, and molecular characteristics of six acute myeloid leukemia (AML) patients.

	Patient 1	Patient 2	Patient 3	Patient 4	Patient 5	Patient 6
**Age at diagnosis**	47 years	66 years	65 years	59 years	62 years	71 years
**Sex**	Male	Male	Male	Female	Male	Female
**WBC/µL at diagnosis**	<100,000	<100,000	<100,000	<100,000	<100,000	>100,000
**WHO classification (2016)**	AML–NOS (Acute myelomonocytic leukemia)	AML with MDS-related changes	NPM1-mutated AML	AML–NOS (AML with maturation)	AML with MDS-related changes	Therapy-related AML
**2017 ELN risk classification**	Adverse	Adverse	Favorable	Adverse	Intermediate	Favorable
**Initial chemotherapy**	aracytin–idarubicin–lenalidomide	aracytin–daunorubicin–selinexor	aracytin–daunorubicin	aracytin–idarubicin	aracytin–idarubicin	cytarabine–daunoubicin–midostaurin
**Consolidation therapy**	ASCT	ASCT	Anti-WT1 vaccination	ASCT	ASCT	Aracytin–midostaurin
**Cytogenetic at diagnosis**	t(6;11), t(11,14), partial tetrazomy of 11q	Hyperdiploidy–trisomy 8	Normal karyotype	Hyperdiploidy–t(2;12), trisomy 4	Normal karyotype	Normal karyotype
**Mutations at diagnosis**	FLT3 and partial MLL tandem duplication	CEBPA–ASLX1–STAG2	NPM1–WT1	WT1–ASLX1–GATA2	/	NPM1–FLT3

WBC: white blood cells; WHO: World Health Organization; AML: acute myeloid leukemia; NOS: not otherwise specified; NPM1: nucleophosmin 1; MDS: myelodysplastic syndrome; ELN: European LeukemiaNet; ASCT: allogeneic stem cell transplantation; t: translocation; FLT3: Fms-like tyrosine kinase 3 gene; MLL: mixed lineage leukemia 1 gene; CEBPA: CCAAT/enhancer-binding protein alpha gene; ASXL1: ASXL transcriptional regulator 1 gene; STAG2: stromal antigen 2 gene; WT1: Wilm’s tumor 1 gene; GATA2: GATA binding protein 2 gene.

**Table 2 diagnostics-10-00317-t002:** Global theoretical validation of the L-Cloud and Abnormality ratio concepts. Six L-Clouds were established on the basis of 10^4 cells from the major malignant cell cluster of six AML patients at diagnosis (patients 1–6). Multiple measurable residual disease (MRD) levels were simulated by bootstrapping (from ≈0.5% MRD to ≈0.001%), using a mixture of normal cells (1 × 10^6^ cells) sampled from 20 reference patients (control group) and adding cells from each patient’s major malignant cell cluster (5000 (≈0.5%), 1000 (≈0.1%), 500 (≈0.05%), 100 (≈0.01%), 50 (≈0.005%), or 10 cells (≈0.001%)). The number of cells retrieved into the patient-specific L-Cloud for the control group and each MRD simulation were determined, and AR was calculated for the MRD simulations. AR: Abnormality Ratio; MRD: measurable residual disease.

		Patient 1	Patient 2	Patient 3	Patient 4	Patient 5	Patient 6
	L-Cloud sensitivity ^‡^	65.75%	66.25%	64.63%	66.76%	68.28%	65.99%
	L-Cloud specificity ^§^	99.9930%	99.9981%	99.9995%	99.9999%	99.9502%	99.9690%
	1–(L-Cloud specificity)	0.0070%	0.0019%	0.0005%	0.0001%	0.0498%	0.0310%
**Number of cells retrieved into the L-Cloud ***	Control group	70	19	5	1	498	310
	0.5% MRD simulation	3234	3228	3791	3567	3948	4037
	0.1% MRD simulation	707	680	773	721	1185	1066
	0.05% MRD simulation	391	345	385	361	841	685
	0.01% MRD simulation	122	85	81	83	566	373
	0.005% MRD simulation	101	48	41	35	531	344
	0.001% MRD simulation	78	26	12	6	506	317
**Abnormality Ratio (AR) ^†^**	0.5% MRD simulation	45.97	169.05	754.43	3549.25	7.89	12.96
	0.1% MRD simulation	10.09	35.75	154.45	720.28	2.38	3.44
	0.05% MRD simulation	5.58	18.15	76.96	360.82	1.69	2.21
	0.01% MRD simulation	1.74	4.47	16.20	82.99	1.14	1.20
	0.005% MRD simulation	1.44	2.53	8.20	35.00	1.07	1.11
	0.001% MRD simulation	1.11	1.37	2.40	6.00	1.02	1.02

* The L-Cloud represents a model describing a multidimensional space where malignant cells are found. This region is patient-specific and is established following a well-defined algorithm at diagnosis (see Section 2.2). † The AR is a calculated parameter obtained by following formula: AR = ((L-Cloud cells per total cells) of simulated sample)/((L-Cloud cells per total cells) of control group sample). ‡ The L-Cloud sensitivity is defined by the number of cells from the “L-Cloud cell cluster” falling into the L-Cloud divided by the number of cells contained into this same cluster (= 10^4^ cells) as a percentage. § The L-Cloud specificity is defined by the number of cells of the control group falling outside the L-Cloud divided by the number of cells of the control group analyzed (= 10^6^ cells) as a percentage.

**Table 3 diagnostics-10-00317-t003:** Comparison of flow cytometry MRD approaches.

Flow Cytometry MRD Approach	LAIP	DfN	AR/L-Cloud
**Type of approach**	Diagnostic-based	Reference-based	Diagnostic-based and reference-based
**Foundations**	Expert knowledge of malignancy (LAIP knowledge)	Knowledge of normality	Unsupervised data analysis (clustering) and comparison to reference samples ^$^
**Subjectivity**	Manual gating	Methodology (unknown *)	Cloud modeling (clustering and contour gating algorithms)
**Standardization**	Impossible	Depends upon methodology	Possible ^$^
**Automation**	Impossible	Depends upon methodology	Possible ^$^
**Reference samples**	Preferable	Required	Required
**Diagnostic sample**	Required	Not required	Required
**Data analysis tools**	Basics	Depends upon methodology	Advanced

LAIP: Leukemia-associated immunophenotype; DfN: different from normal; AR: Abnormality Ratio; L-Cloud: Leukemic Cloud; * to the best of our knowledge; ^$^ considered as an advantage.

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
