# Peer review of "Clustering and Kernel Density Estimation for Assessment of Measurable Residual Disease by Flow Cytometry"

_diagnostics, 2020, doi:10.3390/diagnostics10050317_

Round 1

Reviewer 1 Report

MRD analyses in AML is of growing importance to guide treatment and assess prognosis. Flow MRD and the associated detection/data processing methods are thus of high interest. The manuscript by Jacqmin may contribute. -In general LIAP and DfN approaches both have their advantages and disadvantages; resulting in the recommendation of a combined approach. In general I think it would be helpful to describe - if at all - how the approach with the "LCloud" is different to a LIAP approach and how it would integrate into the methods used today. I also think it must be made clearer what the expected advantages may be from a clinical point of view, this would make the manuscript stronger. -In general, with the given number of patients it is hardly possible to tell how the approach would compare to MRD methods used in everyday clinic (flow or eg NGS, PCR based approaches). It is impossible to tell how well (or not) the method might improve any MRD assessment - a real evaluation from the clinical point of view is impossible. Merely a distribution and a discussion of the approach seems possible. It would have been great to see larger population and even a validation cohort... -Please reconsider the design of figure 1; it may be fancy but makes it hard to read - For the controls, I am still unsure if the subjects were allowed to have a hematologic disease (other then malignancy) after all there must have been a reason to draw BM, rendering them rather unsuitable as controls? Please at least discuss. - Please correct the number formatting (eg 106 or 104) and probably in some of the cases "," should be "." (eg in 0,01% should be be 0.01%) - I also think it is at least worth a discussion, defining a malignant cell by the cells in the L-cloud of AML patients and then drawing conclusions towards sensitivity and specificity. This implies a connection to sensitivity and specificity with the basis of the amount of malignant AML cells detected. But this is something you cannot know. Since biologically not all cells in "the L cloud" may be AML malignant cells and not all cells outside the L cloud need necessarily be non-AML non-malignant cells - For the 6 patients no other MRD assessments were available? Patients' characteristics are not really given. The method of detection for the molecular changes is not given (eg PCR, Sequencing, real/time, NGS..) but may help to evaluate the results. The blasts% are in BM? WT1 is a mutation or expression marker? - For most of the patients the AR was never negative even in patients remaining in remission. Would the authors say that there might be a threshold in the future? May some of the cells in the L cloud be not malignant? - With respect to patient 6 would the others also suggest a combinational approach with DfN in the future? - The supplementary Material was not available to me to review...

Reviewer 2 Report

Current barrier of utilizing multiparameter flow cytometry (MFC) to track residual disease cells comes from the lack of consensus testing method and measurement interpretation. In this paper, Hugues Jacqmin et al developed an algorithm to standardize the analysis of MFC data in AML patients. This method starts from identifying a malignant cell cluster at diagnosis using KNN clustering method. Then, automated two dimensional gates are created using all possible combinations of any two parameters. These contour gates defined the multidimentional region of the Leukemic Cloud (L-Cloud). Finally, an Abnormality Ration (AR) is calculated for each follow-up sample to track the MRD in patients. The authors collected 6 patients to validate their algorithm, and observe good correlation between AR score and the simulated true answer. This work investigated an emerging field of AML, and provided valuable insight. However, some key information about the algorithm and its application is missing. I have the following questions.

  1. How different are the 20 control samples? From the result, it seems there is a large variability in the bootstrapped control cells. Different control cell may lead to different sensitivity and specificity. How to solve this problem in the algorithm standardization?
  2. In table 1, patient 5 and 6 show significant worse outcome compared to the others at low MRD. What are the possible reasons for that? A 4/6 success rate is not ideal. Can we do better?
  3. In Figure 4, the annotation of the arrows is too small to see.
  4. Why patient 2 have two AR score AR1 and AR2?
  5. I see good concordance between AR score and remission. However, the purpose of tracking MRD is to identify relapse ahead of time. But I failed to observe this in the data. If we did not see increase AR score before relapse really occurs, what is the point of doing follow-up flow cytometry?
  6. The AR score varies greatly among patients. How can physicians make the judgement of a potential relapse based on your AR score?
  7. Line 124. The number of cells 1x106 show be power of 6, not 106. Similar typo in line 129 and 135.

Round 2

Reviewer 2 Report

My concerns have been addressed.

One last comment, I suggest the algorithm and example data can be open-sourced at publication so other scientists can use them. That would be very beneficial to the research community.
